# Osteosarcopenia in NAFLD/MAFLD: An Underappreciated Clinical Problem in Chronic Liver Disease

**DOI:** 10.3390/ijms24087517

**Published:** 2023-04-19

**Authors:** Alessandra Musio, Federica Perazza, Laura Leoni, Bernardo Stefanini, Elton Dajti, Renata Menozzi, Maria Letizia Petroni, Antonio Colecchia, Federico Ravaioli

**Affiliations:** 1Department of Medical and Surgical Sciences, IRCCS-Azienda Ospedaliero-Universitaria di Bologna, 40138 Bologna, Italy; alessandra.musio@studio.unibo.it (A.M.); federica.perazza@studio.unibo.it (F.P.); laura.leoni@studio.unibo.it (L.L.); bernardo.stefanini@studio.unibo.it (B.S.); elton.dajti2@unibo.it (E.D.); marialetizia.petroni@unibo.it (M.L.P.); 2Division of Metabolic Diseases and Clinical Nutrition, Department of Specialistic Medicines, University Hospital of Modena and Reggio Emilia, Largo del Pozzo 71, 41125 Modena, Italy; renata.menozzi@unimore.it; 3Gastroenterology Unit, Department of Medical Specialties, University Hospital of Modena, University of Modena & Reggio Emilia, 41121 Modena, Italy; acolecch@unimore.it

**Keywords:** non-alcoholic fatty liver disease, non-alcoholic steatohepatitis, metabolic dysfunction-associated fatty liver disease, cirrhosis, advanced chronic liver disease, sarcopenia, vitamin D, osteoporosis, nutrition, chronic liver disease

## Abstract

Chronic liver disease (CLD), including non-alcoholic fatty liver disease (NAFLD) and its advanced form, non-alcoholic steatohepatitis (NASH), affects a significant portion of the population worldwide. NAFLD is characterised by fat accumulation in the liver, while NASH is associated with inflammation and liver damage. Osteosarcopenia, which combines muscle and bone mass loss, is an emerging clinical problem in chronic liver disease that is often underappreciated. The reductions in muscle and bone mass share several common pathophysiological pathways; insulin resistance and chronic systemic inflammation are the most crucial predisposing factors and are related to the presence and gravity of NAFLD and to the worsening of the outcome of liver disease. This article explores the relationship between osteosarcopenia and NAFLD/MAFLD, focusing on the diagnosis, prevention and treatment of this condition in patients with CLD.

## 1. Introduction

Non-alcoholic fatty liver disease (NAFLD), also recently named metabolic dysfunction-associated fatty liver disease (MAFLD), is the most frequent chronic liver disease—a non-infectious epidemic which burdens the healthcare system, NAFLD/MAFLD affects about 25% of the world adult population. NAFLD is increasingly considered given the involvement of the liver in metabolic syndrome. A survey reported that NAFLD is the only systematically increasing liver disease in the USA over the past three decades, associated with increased obesity and type 2 diabetes mellitus (T2DM). Furthermore, in the past two decades, the prevalence of NAFLD has increased in the Asian population due to the spread of a sedentary lifestyle, being overweight and T2DM. From histological findings, NAFLD is described as a disproportionate fat accumulation in hepatocytes, combined with insulin resistance, and is defined by steatosis in >5% of liver cells without evidence of significant hepatocyte ballooning indicative of hepatocellular damage in the absence of competing liver disease aetiologies. NAFLD may worsen and cause non-alcoholic steatohepatitis (NASH), defined by hepatic steatosis and inflammation with hepatocyte ballooning; it is possible to observe further progression to hepatic fibrosis, cirrhosis and hepatocellular carcinoma.

Osteoporosis is a bone metabolism disease in which bone becomes more fragile, and the risk of fracture following a small trauma increases since the mineral density is reduced. Sarcopenia, such as osteoporosis, is a frequent condition in the elderly and is often associated with NAFLD; sarcopenia is characterised by progressive and systemic reduction in skeletal muscle mass, strength or function, with an increased risk of disability, hospitalisation and mortality. These two conditions share several common pathophysiological pathways, among which insulin resistance; chronic systemic inflammation, particularly increased levels of inflammatory cytokines, i.e., IL-1, IL-6 and TNF-α; and physical inactivity represent crucial pathophysiological elements and promote muscle and bone loss; moreover, they predict the presence and severity of NAFLD and worsen the outcome of liver disease. Interestingly, Semaglutide and Liraglutide, anti-diabetic and anti-obesity drugs, improve liver steatosis and inflammation and at the same time seem to have a positive effect on bone. In this review article, we aim to summarise the current evidence on the impact of sarcopenia and bone impairment in NAFLD/MAFLD patients and the strategy for the early identification and management of these patients. Their evaluation, management and treatment should be more appreciated in clinical practice in NAFLD patients.

## 2. Data Sources and Searches

We searched English-language publications in MEDLINE, Ovid, In-Process, the Cochrane Library, EMBASE, and PubMed until December 2022. Literature searches were performed using the following keywords: non-alcoholic fatty liver disease, NAFLD, metabolic dysfunction-associated fatty liver disease, MAFLD, chronic liver disease, advanced chronic liver disease, sarcopenia, cachexia, malnutrition, bone impairment, osteopenia and osteoporosis.

## 3. Sarcopenia

### 3.1. Definition of Sarcopenia (S) and Sarcopenic Obesity (SO)

Sarcopenia has been defined by the European Working Group on Sarcopenia in Older People as a progressive and generalised skeletal muscle disorder that increases the likelihood of adverse outcomes, including falls, fractures, physical disability and mortality [1]. According to the consensus, sarcopenia is probable when low muscle strength is detected, whereas low muscle quantity or quality confirms sarcopenia diagnosis. When low muscle strength, low muscle quantity/quality and low physical performance are all detected, sarcopenia is considered severe [1]. In clinical practice, SARC-F, a five-item questionnaire answered by patients, is recommended to screen for sarcopenia risk [2]. Table 1 summarizes the diagnostic methods proposed.

Muscle strength should be evaluated by handgrip strength (HGS) or the chair stand test (CST) [1]. HGS records the mean value (in kilograms) of three consecutive measurements of the dominant arm gripping a dynamometer [1]. The CST measures the time the patient needs to rise five times from a seated position without using their arms [1].

Low muscle strength is defined as handgrip strength (HGS) < 27 kg in men or <16 kg in women, or as a time > 15 s for five rises in the chair stand test.

To confirm the diagnosis of sarcopenia, we need to assess an alteration in muscle quantity or quality. According to the consensus mentioned above [1], magnetic resonance imaging (MRI) and computed tomography (CT) are considered to be the gold standards for non-invasive assessment of muscle quantity/mass, but dual-energy X-ray absorptiometry (DXA) and, to a lesser extent, bioelectrical impedance analysis (BIA) are mentioned as good and reliable methods to assess muscle mass [1].

CT allows quantification of the skeletal muscle area (SMA) in cm^2^. SMA is then adjusted to the squared height to obtain the skeletal muscle index (SMI) in cm^2^/m^2^. The lumbar skeletal muscle area correlates relatively well with the whole body muscle mass, especially at the third lumbar vertebra (L3) [3,4].

BIA is based on the whole-body electrical conductivity of hydrated tissues, allowing estimation of lean body mass through instrument-specific and population-specific regression equations [5]. Moreover, it measures phase angle (PA), which estimates cellular membrane integrity and cellular water distribution without using a prediction equation [5].

DXA offers further advantages since it measures mass, density, total body and appendicular fat and fat-free mass [6].

When assessed by BIA or DXA, muscle quantity can be reported as total body skeletal muscle mass (SMM) or appendicular skeletal muscle mass (ASM).

Low muscle quantity is defined as appendicular skeletal mass (ASM) < 20 kg for men and <15 kg for women, or as ASM/height^2^ < 7.0 kg/m^2^ for men and <5.5 kg/m^2^ for women.

The term sarcopenia is often used to refer only to a reduction in muscle mass, based on the sole criterion of low muscle area and utilising the skeletal muscle index (SMI): the total abdominal skeletal muscle area in cm^2^, measured at the upper level of the third lumbar vertebra (L3), normalised to height (cm^2^/m^2^). Sarcopenia is defined when the skeletal muscle index (SMI) is <41 cm^2^/m^2^ for a woman and <53 cm^2^/m^2^ for a man in the field of obesity [7], or <39 cm^2^/m^2^ for a woman and <50 cm^2^/m^2^ for a man in the field of end-stage liver disease (ESLD) [8].

Recently, in a joint Consensus Statement from the European Society for Clinical Nutrition and Metabolism (ESPEN) and the European Association for the Study of Obesity (EASO), sarcopenic obesity was defined as the co-existence of obesity and sarcopenia, diagnosed by altered skeletal muscle functional parameters and altered body composition [9]. The expert panel suggested adopting the cut-points proposed by Dodds et al. [10] (for the Caucasian population) and Chen et al. [11] (for the Asian population) for HGS and the references given by Gallagher et al. for fat mass (FM) [12], by Janssen et al. for SMM/weight [13] and by Batsis et al. for appendicular lean mass/weight (ALM/W) [14].

Myosteatosis can be considered a qualitative alteration of muscle due to fat accumulation in the muscle [15]. It is associated with an increased prevalence of NAFLD [16], with the presence and severity of NASH [17] and significant fibrosis [18].

In a cohort of 338 Korean patients with biopsy-proven NAFLD, the prevalence of severe myosteatosis was 21.1% and 33.3% in the NAFLD and early NASH groups, respectively (*p* = 0.029) [19].

In a study including 675 patients with liver cirrhosis (LC) evaluated for liver transplant (LT) (23% with NASH-cirrhosis), myosteatosis was present in 52% [20]. Furthermore, in a study enrolling 265 patients listed for LT, among the 136 patients with NASH, 62% had myosteatosis [21]. In a cohort of 678 patients with LC, 77 (11.4%) had both myosteatosis and sarcopenic obesity [22].

### 3.2. The Prevalence of Sarcopenia in NAFLD and Cirrhotic-NASH and Prognosis Implication

#### 3.2.1. Sarcopenia and NAFLD

The prevalence of sarcopenia is significantly increased in NAFLD, ranging from 12.2% to 43.6% [23,24,25,26,27,28,29], and in NASH (about 35%) [23], compared to that in non-NAFLD patients, ranging from 8% to 9.7% [23,26]. On the other hand, some studies have identified a low prevalence of sarcopenia in NAFLD patients, ranging between 3.5% [30] and 4.5% [31]. This high heterogeneity is probably due to the differences among studies in the definition and assessment of sarcopenia, NAFLD and NASH.

Several studies have pointed out the association between NAFLD and sarcopenia. A recent meta-analysis by Cai et al. [32] included 19 studies, of which 17 were cross-sectional and only 2 were retrospective cohort studies [33,34]. The authors observed a significantly higher occurrence risk of NAFLD (OR = 1.33, 95% CI 1.20 to 1.48), NASH (OR = 2.42, 95% CI 1.27 to 3.57) and NAFLD-related significant fibrosis (OR = 1.56, 95% CI 1.34, 1.78) in subjects with sarcopenia [32].

In the two retrospective cohort studies (with 10-year [34] and 7-year follow-ups [33]), skeletal muscle mass was inversely associated with the incidence of NAFLD [33,34] and positively associated with its resolution [33].

Additional cross-sectional and retrospective studies have further confirmed the independent association between sarcopenia and NAFLD [35,36,37,38,39,40,41,42], NAFLD severity [35,37,43], fibrosis [36,40,44,45,46,47] and fibrosis severity [36,48,49].

Even if limited, prospective studies assessing the relationship between sarcopenia and NAFLD have also been conducted. In a prospective study enrolling 156 consecutive patients with biopsy-proven NAFLD and alanine aminotransferase (ALT) > 40 IU/L, 13.5% and 26.3% were diagnosed with sarcopenia with the SMI and sarcopenia index (SI; ASM-to-body mass index (BMI) ratio), respectively; in patients with hepatic fibrosis stage < 2, the SI and the skeletal muscle mass-to-body fat mass ratio were significantly higher than in patients with fibrosis stage ≥ 2 [50]. In another prospective study including more than 300,000 participants with a median follow-up of 10 years, lower muscle mass and grip strength were associated with a higher risk of developing severe NAFLD [51].

Furthermore, sarcopenia in NAFLD patients has been associated with adverse clinical outcomes, including a higher 10-year major osteoporotic and hip fracture probability [52], increased risk of albuminuria [53] and atherosclerotic cardiovascular disease (ASCVD) [46,54,55], and increased risk of all-cause [25,56,57,58,59], cardiovascular [25,57], diabetes-associated [25,58] and cancer-associated [58] mortality.

Conversely, some studies failed to show an association between low muscle mass and NAFLD [31] while identifying fat mass, particularly the android fat mass and the android-to-gynoid fat ratio, as a better predictor for NAFLD [31].

These studies had several limitations. First, different methods were used to assess skeletal muscle mass, such as bioelectrical impedance analysis (BIA) and dual-energy X-ray absorptiometry (DXA), with a few studies utilising computed tomography (CT). Furthermore, the definition of SMI varies among different studies. Most studies defined SMI as skeletal muscle mass (SMM) divided by body weight or BMI, while only a few divided SSM by height. Therefore, the definition of sarcopenia is influenced by the patient’s weight, and a higher BMI may have a greater impact on the development of NAFLD than low muscle mass per se. It should be noted that, in the ESPEN and EASO consensus on sarcopenic obesity, the authors point out that a relative reduction in muscle mass in patients with high BMI and FM may have a relevant clinical and functional impact even in the absence of an absolute muscle mass reduction [9].

Adjusting muscle mass for BMI or height can lead to divergent results when evaluating the association between sarcopenia and NAFLD, as shown in a cross-sectional study including 320 participants [60].

In this study, muscle mass adjusted for BMI was associated with NAFLD diagnosed by ultrasound (US; OR, 1.71; 95% CI, 1.02 to 2.86) and comprehensive NAFLD score (CNS) (OR, 1.95; 95% CI, 1.04 to 3.65), whereas muscle mass adjusted for height was not associated with NAFLD [60].

The use of liver biopsy to evaluate NAFLD and hepatic fibrosis is another area for improvement with respect to existing studies, since this is available in only a few of them.

Third, because most studies focused on the Asian population, the findings are not widely applicable.

Altered muscle quality, identified as lower muscle attenuation or density, has been associated with prevalence of NAFLD [16], development of NASH [19,59], liver stiffness [18] and fibrosis progression [19]. In research including 9545 participants, the participants with NAFLD and adverse muscle composition (AMC), defined as low muscle volume and high muscle fat, had a 2.1-fold and 3.3-fold higher prevalence of type 2 diabetes and coronary heart disease (*p* < 0.001), respectively, compared with those without AMC [27]. On the other hand, in a cross-sectional study of 45 NAFLD patients, the muscle fat fraction (MFF) was not correlated with the hepatic fat fraction (r = −0.035, *p* = 0.823) and did not significantly differ between subjects with or without significant fibrosis [61].

Studies have also examined the relationship between NAFLD patients’ muscle strength and physical performance assessment. Cross-sectional studies have detected an association between lower muscle strength, NAFLD and liver fibrosis severity [37,62,63,64,65,66], and the association between SMM and NAFLD has also been investigated in children and adolescents. In a case–control study including 53 paediatric patients with NAFLD and 73 controls aged 9–15 years, the skeletal muscle-to-body fat ratio, assessed by BIA, was significantly lower in individuals with NAFLD than in those without (0.83 vs. 1.04, *p* = 0.005), even after adjusting for age, sex, BMI and serum glucose [67]. Low muscle mass was associated with an increased risk for NAFLD in overweight/obese youths [68]. Furthermore, SMI and AMI were negatively associated only with steatosis in obese adolescents, without being associated with NAFLD activity score (NAS), lobular inflammation, ballooning scores or fibrosis stage [69].

#### 3.2.2. Sarcopenia and NASH Liver Cirrhosis (LC)

Sarcopenia prevalence in patients with liver cirrhosis (LC) of different aetiologies (prevalence of cirrhotic NASH not specified) varies between 21.7% and 76% [70,71,72,73,74,75,76,77,78]. In studies involving patients with LC of different aetiologies and with a prevalence of NASH/NAFLD ranging from 9.8% to 34.5%, sarcopenia prevalence varies from 19.8% to 48% [79,80,81]. In a meta-analysis of 22 studies involving 6965 patients [82], the pooled prevalence of sarcopenia in patients with LC was 37.5% overall (95% CI 32.4–42.8%) and was higher in males, those with alcohol-associated liver disease, those with Child–Pugh grade C and when sarcopenia was defined by L3-SMI (i.e., assessed with the third lumbar skeletal muscle index) [82,83], independently of the model for end-stage liver disease score (MELD) [83]. A meta-analysis of 20 studies (7 Asian and 13 Western) estimated a prevalence of sarcopenia in LC patients of 48.1%, with a higher rate in men (61.6%) than in women (36%) [84].

Sarcopenia prevalence appears to increase along the different Child–Pugh stages (CPS) (18.2% in CPS A, 42.4% in CPS B and 90.5% in CPS C) [85], and after autoimmune hepatitis-related cirrhosis (80%), NASH-related LC is highest (61.9%) [85].

In patients with LC, sarcopenia has been associated with reduced quality of life [85,86]; increased risk of LC complications, such as ascites [76,87,88,89,90], hepatic encephalopathy (HE) [20,87,88,91,92,93,94,95,96,97,98,99], HCC [76,100,101,102], variceal bleeding [88], spontaneous peritonitis [76], infections [76], more extended hospital stays [76,103,104], higher hospital costs [103,104], higher 30-day readmission [76] and overall increased mortality [78,82,89,92,103,104,105,106,107,108,109,110,111,112]; as well as reduced survival [76,81,113,114,115,116,117]. As shown in a meta-analysis including 20 studies, LC patients with sarcopenia had poorer survival rates and an increased risk of complications, such as infection, than those without sarcopenia [84].

Moreover, myosteatosis has been associated with higher mortality in LC patients [22,107,118] and with the risk of developing HE in LC patients [20,119], but not with the length of hospital stay in patients with decompensated cirrhosis [120].

Recently, when body composition features, such as sarcopenia and myosteatosis, were integrated with the model for end-stage liver disease (MELD) [121,122,123,124] and Child–Turcotte–Pugh score (CTP) [125], they increased the accuracy of this model in predicting mortality [121,122,123,124] and hospital readmission [125], particularly in those with compensated advanced chronic liver disease [126].

#### 3.2.3. Sarcopenia and Liver Transplant (LT)

In a meta-analysis including 3.803 patients on waiting lists (WLs) or undergoing liver transplantation (LT), van Vugt et al. [83] reported a sarcopenia prevalence ranging from 22% to 70% [83]. Sarcopenia has been associated with WL [83,127,128], post-LT mortality [83,129,130,131,132] and with post-LT complications [73,130,131,133,134]. Only a few studies found no association between sarcopenia and post-LT mortality [73,134,135].

In patients with LC of different aetiologies, the prognostic value of myosteatosis seems to be particularly important in the early post-operative phase, with higher rates of deaths due to respiratory and septic complications [136]. Furthermore, low muscle quality was a statistically significant predictor of an increased risk for WL mortality, with an HR of 9124 (95% CI 2871–28,970) [128].

Most studies regarding the influence of sarcopenia on transplant-related outcomes involved patients with different aetiologies of chronic liver disease (CLD). In a study including 265 patients evaluated for their first LT, of whom 126 had a primary diagnosis of NASH and 129 of alcoholic liver disease (ALD), patients with NASH had a significantly lower prevalence of sarcopenia (22% versus 47%; *p* < 0.001) when compared with patients with ALD, which was not associated with LT complications [21]. A retrospective single-centre study including 146 adult patients who received LT for NASH-LC, which evaluated the association between sarcopenia and clinical outcomes, did not observe any significant difference between patients with or without sarcopenia in the length of hospitalisation following LT, re-hospitalisation in the first year post-LT, or one-year or overall survival [137].

#### 3.2.4. Sarcopenic Obesity and NAFLD

Several studies have related sarcopenic obesity (SO) to NAFLD. In a multicentre, retrospective study involving 23,889 NAFLD subjects, SO prevalence was 12.0% (2872 pts.) [138].

However, the current evidence shows high heterogeneity in the SO definitions, including increased waist-to-calf ratio [139], reduced skeletal muscle mass-to-visceral fat area ratio (SV ratio) [140,141,142], increased visceral fat area-to-appendicular muscle mass ratio (VAR) [143] or reduced appendicular skeletal muscle mass-to-visceral fat area ratio [144], the combination of low skeletal muscle mass or myosteatosis with visceral adiposity [48], and the combination of low muscle mass and strength with obesity defined in terms of BMI or waist circumference [39]. Surrogate markers of sarcopenic obesity were independently associated with a higher risk of NAFLD [39,139,141,143], NASH [144] and significant fibrosis [48,139] than the two components (sarcopenia and obesity) alone [39]. In a retrospective study involving 92 patients with NAFLD followed up for a median of 4.1 years, patients with a worsened SV ratio had higher liver stiffness and fat accumulation [140]. In a post hoc analysis of the ATTICA study (2020 patients completed the follow-up), participants with low SMI and abnormal waist circumference exhibited the highest NAFLD rate compared to those with moderate/high SMI and normal waist circumference (60.5% and 24.3%, respectively; *p* < 0.001) [55].

In a few studies, SO was diagnosed as a combination of low muscle mass and an increased percentage of fat mass [145,146]. SO was associated with lean NAFLD [145], a higher risk of NAFLD [146] and significant liver fibrosis [146].

In a multicentre, retrospective study involving 23,889 NAFLD subjects, Chun et al. [138] validated a model of high-risk and low-risk SO, which included SI (total ASM (kg)/body mass index), age, sex, and presence or absence of metabolic syndrome. After full adjustment, high-risk SO subjects had a significantly higher risk for significant liver fibrosis or atherosclerotic cardiovascular disease (ASCVD). After a median follow-up of 3 years, the cumulative incidences of significant liver fibrosis (F ≥ 2), CVD, cirrhosis and all-cause mortality were significantly higher in high-risk SO subjects [138].

#### 3.2.5. Sarcopenic Obesity and LC

Studies have reported an SO prevalence in LC of 20% to 35% [22,115,147,148,149,150].

Among LC aetiologies, NASH has been established as an independent predictor of SO [148].

As reported in a meta-analysis including retrospective cohort studies and 1315 pre-LT patients (with a percentage of NASH as aetiology ranging from 4% to 22%), SO prevalence varied from 13% to 33% [151]. Moreover, SO increased overall mortality compared to non-SO at short- (1 year), intermediate- (3 years) and long-term follow-up (5 years), with RRs of 2.06 [95% CI: 1.28–3.33], 1.67 [95% CI: 1.10–2.51] and 2.08 [95% CI: 1.10–3.93], respectively [151]. In other studies involving patients with LC, sarcopenic obesity worsened the prognosis in Child–Pugh A patients [115] and increased post-transplant mortality [152].

**Table 1 ijms-24-07517-t001:** Methods to assess sarcopenia in patients with chronic liver disease and NAFLD.

Methods	Procedure	Units	Cut-Offs Proposed in NAFLD	Cut-Offs in LC	Pros	Cons
SARC-F	5-item questionnaire self-reported by patients	NA	NS	≥4 [153]	Inexpensive; simple; repeatable; high specificity	Subjective Low sensitivity
HGS	Records the mean value of three consecutive measurements of the dominant arm gripping a dynamometer	kg	M: ≤29 for BMI ≤ 24 kg/m^2^, ≤30 kg for BMI 24.1–28 kg/m^2^, ≤32 kg for BMI > 28 kg/m^2^ F: ≤17 kg for BMI ≤ 23 kg/m^2^, ≤17.3 kg for BMI 23.1–26 kg/m^2^, ≤18 kg for BMI 26.1–29 kg/m^2^, ≤21 kg for BMI >29 kg/m^2^ [31] M: <26; F: <18 [26,154]	M: <26 F: <18 [155]	Simple and inexpensive High sensitivity	Low specificity Not representative of overall sarcopenia
SMI by CT	Total abdominal skeletal muscle area in cm^2^, measured at the upper level of the third lumbar vertebra (L3), normalised to height	cm^2^/m^2^	M: <50 F: <39 [48] Expressed as total muscle area/BMI M:<8.37 F: <7.47 [156]	M: <50 F: <39 [8] M: <42 F: <38 [155] M: <44.77 F: <32.50 [75]	Accurate; rapid	Radiation exposure, not available at the bedside, varying cut-points/sites of measurement and not easily repeatable
PMA	The sum of the areas of the two psoas at the level of the third or fourth lumbar vertebra	mm^2^	/	M: <1561 F: <1464 [114]	Accurate; no specific software is needed	Not representative of overall sarcopenia
PMI	Total bilateral psoas muscle area at the middle of the third lumbar vertebra (L3) level (cm^2^), shown by CT and height (m)	cm^2^/m^2^	/	M: <5.1 F: <4.3 [157] M: <5.16 F: <4.54 [125]	Simple and commonly used	Not representative of overall sarcopenia
PDI	The ratio of the measured PMA to the estimated peak PMA, which represents the peak value of the PMA in healthy individuals	/	/	<0.75	Accurate; no specific software is needed	Not representative of overall sarcopenia
PMTH	Psoas mass thickness, measured on CT at the level of the umbilicus, normalised by division by height	mm/m	NS	M: <17.3 F: <10.4 [158]	Easy to calculate	The level of the umbilicus may vary if ascites is present Not representative of overall sarcopenia
MRA	Assessed as the mean density (HU) of the entire measured cross-sectional muscle area at L3, measured on CT	HU	<42.57 HU if BMI ≥ 25 <39.77 if BMI < 25 [48]	M: <33 F: <28 [118]	Indicative of muscle quality	No consensus regarding use in clinical practice
FFMA	Total erector spinae muscle area and the intramuscular fat tissue measured and subtracted on MRI at the level of the radix of the superior mesenteric artery	mm^2^	/	M: <3197 F: <2895 [159]	Indicative of muscle quality Easy to identify Very low inter-reader variability	No consensus regarding use in clinical practice High cost Limited availability
TPMT	Greatest transverse diameter of the right psoas muscle perpendicular to the long axis (anterior–posterior oblique) of the psoas muscle diameter at the cranial L3 vertebra endplate, measured by MRI and normalised by height	mm/m	/	M: <12 F: <8 [109]	Rapid Reproducible Excellent inter- and intra-reader agreement No contrast needed	High cost Limited availability Further validation needed
SMI by DEXA	Percentage of total skeletal muscle mass obtained by DXA, normalised by weight	%	M: <39.8 F: <34.1 [160]	/	Easy; reproducible; whole body	Radiation Influenced by water retention
ASM	Summing of the muscle mass of the upper and lower limbs obtained by DXA and normalised by height	Kg/m^2^	M: ≤7 F: ≤5.4 [26,145,154] M: ≤7.25 F: ≤5.67 [31,62]	M: <7.26 F: <5.45 [70,161,162] M: <7 (+HGS < 25 kg) [163] M: <7 F: <5.5 [164]	Easy; reproducible	Radiation Water retention may cause an overestimation of the fat-free mass
ASM (%)	Summing of the muscle mass of the upper and lower limbs obtained by DXA and normalised by weight	%	M: <32.2 F: <25.5 [165]		Easy; reproducible	Radiation Water retention may cause an overestimation of the fat-free mass
SI	Summing of the muscle mass of the upper and lower limbs obtained by DXA and normalised by BMI		M: <0.789 F: <0.521 [29] M: <0.789 F: <0.512 [25] M: <0.678 F: <0.468 [166]		Easy; reproducible	Radiation Water retention may cause an overestimation of the fat-free mass
ULMMI	Muscle mass of the upper limb obtained by DXA and normalised by height	Kg/m^2^		M: <2.1 F: <1.5 [167]	Easy, reproducible and less influence on water retention	Radiation
SMI by BIA	Summing of the muscle mass of the upper and lower limbs obtained by BIA and normalised by weight	%	M: ≤30.6 [168] M: ≤37; F: ≤28 [28,36,169] M: ≤30; F: ≤26.8 [170] M: <29; F: <22.9 [23,45] M: <29.1; F: <23 [171]		Safe, rapid, widely available, minimal to moderate training, repeatable and portable	Influenced by hydration status and by weight; estimates are instrument- and population-specific
SMI by BIA	Summing of the muscle mass of the upper and lower limbs obtained by BIA and normalised by height	Kg/m^2^	M: <7.0 F: <5.7 [50,172] M: <10.76 F: <6.75 [169]		Safe, rapid, widely available, minimal to moderate training, repeatable and portable	Influenced by hydration status and by weight; estimates are instrument- and population-specific
SI by BIA	Summing of the muscle mass of the upper and lower limbs obtained by BIA and normalised by BMI	/	M: <0.789 F: <0.512 [45,171] M: <0.789 F: <0.521 [50]		Safe, rapid, widely available, minimal to moderate training, repeatable and portable	Influenced by hydration status and by weight; estimates are instrument- and population-specific
PA by BIA	The ratio of resistance (intracellular and extracellular resistance) to reactance (cell-membrane-specific resistance) expressed as an angle	°	≤5.05 [173] M: ≤5.6 F: ≤5.4 [174]		Good reliability, even in patients with fluid retention	
Right thigh muscle thickness by US	Marking of points 1/3 and 1/2 of the total distance from the top of the patella to the iliac crest. Two readings at each point: one with compression of the probe and the other without; both points averaged and corrected for height	cm/m^2^	/	/	Safe, rapid, accessible and repeatable Used to assess both muscle quantity and quality	Operator-dependent No established cut-offs
Rectus abdominis thickness by US	The thickness of each rectus abdominis at about 3 cm laterally from the umbilicus, between the anterior and posterior fascial borders, applying minimal pressure	mm	/	/	Safe, rapid, accessible and repeatable Used to assess both muscle quantity and quality	Operator-dependent No established cut-offs
Psoas muscle thickness by US	Maximum distance between the anterior and the posterior borders of the psoas muscle, perpendicular to the longitudinal fibres at a level slightly above the iliac crest	mm	/	/	Safe, rapid, accessible and repeatable Used to assess both muscle quantity and quality	Difficult identification because of bowel gas and ascites Operator-dependent No established cut-offs
US-PTHR	Average of three measurements of the largest right psoas muscle diameter divided by height	mm/m	/	/	Safe, rapid, accessible and repeatable Used to assess both muscle quantity and quality	Operator-dependent No established cut-offs
US-PMI	π × psoas radius square divided by the patient’s height square	cm^2^/m^2^	/	/	Safe, rapid, accessible and repeatable Used to assess both muscle quantity and quality	Operator-dependent No established cut-offs

BIA, bioelectrical impedance analysis; HGS, handgrip strength; HU, Hounsfield units; MRA, muscle radiation attenuation; MRI, magnetic resonance imaging; NA, not applicable; NS, specific cut-offs not available; PA, phase angle; PDI, psoas depletion index; PMI, psoas muscle index; SMI, skeletal muscle index; SMM, skeletal muscle mass; LBM, total lean body mass; PMTH, psoas muscle thickness by height; US-PTHR, ultrasound psoas-to-height ratio; US-PMI, ultrasound psoas muscle index.

### 3.3. Diagnosis and Clinical Assessment of Sarcopenia in NAFLD and LC

SARC-F, the most-used tool to screen for sarcopenia risk, has low sensitivity and positive predictive value in patients with CLD [153], but when combined with HGS and the calf circumference or finger-circle test (i.e., the determination of whether or not the maximum non-dominant calf circumference is bigger than the individual’s finger-ring circumference, which is formed by the thumb and forefinger of both hands) it is a valuable screening method for sarcopenia [153]. As regards the assessment of muscle strength, contrary to the CST, low HGS has been associated with NAFLD presence [37,63,64,65,66] and severity [37,63,64] and with the presence of advanced liver fibrosis [65]. Furthermore, low HGS predicts a poor clinical outcome [175] and mortality [161,176] in patients with LC. Therefore, as suggested by De et al. [177], because of its high sensitivity and low specificity, HGS may be used as a screening to stratify CT-based assessment of sarcopenia [177]. In 2016, the Japan Society of Hepatology suggested cut-offs of under 18 kg for women and 26 kg for men with LC [155].

As mentioned above, CT is the gold standard for assessing liver disease muscle quantity. It can distinguish between fluids and soft tissues [178], such as MRI [179]; therefore, it is not influenced by ascites or fluid overload, which is common in patients with LC. On the other hand, both BIA [180] and DXA [181,182] are affected by fluid overload. Furthermore, CT is a routine part of LC follow-up in most centres to screen for HCC [183].

Different definitions of mass muscle reduction have been proposed. Carey et al., in a multicentre American study in patients with end-stage liver disease, suggested SMI cut-offs of <50 cm^2^/m^2^ for men and <39 cm^2^/m^2^ for women based on optimal correlation with survival outcomes [8]. These cut-offs have been incorporated into the European Association for the Study of Liver (EASL) Clinical Practice Guidelines on nutrition in chronic liver disease [184]. Unfortunately, these criteria may not apply to all ethnic groups [185,186]. Therefore, the Japan Society of Hepatology has successively proposed different cut-offs: 38 cm^2^/m^2^ for women and 42 cm^2^/m^2^ for men [155]. Zeng et al. [75] have established the following cut-offs for Chinese LC patients: 44.77 cm^2^/m^2^ in male patients and 32.50 cm^2^/m^2^ in female patients.

Some authors have suggested measuring the dimensions and surface area of the major psoas muscle since this does not require dedicated software. The proposed measures for patients with LC include psoas muscle area (PMA), the sum of the areas of the two psoae at the level of the third or fourth lumbar vertebra, with a cut-off of 1561 mm^2^ in men and 1464 mm^2^ in women [114]; the psoas muscle thickness per height (PMTH) at the umbilical level, with a cut-off of 17.3 mm/m in men and 10.4 mm/m in women [158]; and the psoas depletion index (the ratio of the measured PMA to the estimated peak PMA, which represents the peak value of the PMA in healthy individuals), with a cut-off value of 0.75 [187].

Unfortunately, some authors have pointed out the poor performance of the psoas muscle index (PMA adjusted to the squared height—PMI) in identifying patients with increased mortality risk [157].

Altered muscle quality can also be identified through CT; muscle radiation attenuation (MRA) expressed in Hounsfield units (HU) measures muscle quality, which is inversely related to muscle fat content and quality. The proposed cut-offs to identify myosteatosis for the density of the cross-sectional muscle area at L3 are less than 33 and 28 HU in males and females, respectively [118].

MRI is often available in patients with LC, too. Indeed, it is increasingly used in CLD for specific medical scenarios, such as accurately detecting hepatocellular carcinoma (HCC) [188]. Furthermore, MRI is considered an interesting tool for evaluating body composition because of the lack of radiation exposure and the possibility of obtaining high-quality images, including information on muscle quality as evidenced by fat infiltration [189].

L3 single-slice imaging was the most accurate method used to estimate full-body skeletal muscular area and adipose tissue via MRI [190], with excellent concordance with CT [191]. In a retrospective single-centre study including 265 patients with CLD, sarcopenia was defined by height-adjusted and gender-specific cut-offs as transverse psoas muscle thickness <8 mm/m in women and <12 mm/m in men [109]. MRI has also been investigated to detect muscle quality, evaluated as fat-free muscle area (the subtraction of the intramuscular fat tissue area from the total erector spinae muscle area) [159] or as an apparent diffusion coefficient [192].

Whole-body dual-energy X-ray absorptiometry (DXA) represents a method with lower cost and radiation than CT [193]. DXA cannot be used to assess muscle mass repeatedly since it exposes patients to ionising radiation. Some authors have pointed out a weak correlation between DXA and CT in LC patients [70,161]. Its major limitation is the tendency to underestimate sarcopenia in the case of fluid retention, which is common in patients with LC. Therefore, Belarmino et al. [163] proposed the measurement of appendicular skeletal muscle index (ASMI), which was not influenced by ascites or lower limb oedema [163]. In other studies, lean arm mass used to reduce the effect of lower limb oedema was superior to ASMI in predicting mortality in LC patients [162,164,167].

The use of BIA in decompensated LC is controversial because of the possible overestimation of muscle mass due to extracellular fluid overload. Therefore, researchers have tried to develop methods to assess muscle mass that are radiation-free and can be easily performed at the patient’s bedside, such as ultrasonography (US) and BIA. In NAFLD, sarcopenia, assessed by BIA, is defined as a skeletal muscle mass index of ≤37 in males and ≤28 in females [28], which correlates with clinical outcomes in adults [28] and with MRI in children [194]. Otherwise, in patients with LC, the proposed cut-offs for SMI are <7.0 cm^2^/m^2^ for males and <5.7 cm^2^/m^2^ for female patients [172]. Pirlich et al. [195] highlighted a low agreement between body cell mass (BCM) estimated by BIA and BCM derived from total body potassium counting, which was the gold standard for assessing body composition [195]. Subsequently, Luengpradidgun et al. [196] compared BIA with CT, showing a fair correlation between the two measures (r = 0.54; *p* < 0.002).

On the other hand, in a study involving 122 male patients with LC, phase angle (PA) values ≤ 5.05° were able to predict the diagnosis of sarcopenia with high sensitivity [173]. Furthermore, in a study by Ruiz Margàin et al. [174], PA was independently associated with mortality in compensated cirrhosis, and its prognostic accuracy was not influenced by the presence of ascites [174].

The US allows the measurement of cross-sectional areas, muscle thickness and echo intensity. US measures, such as right thigh muscle thickness [197], rectus abdominis thickness (RA) [198] and psoas muscle diameter-to-height ratio [199], have been correlated with CT diagnosis [197], survival [198] and mortality [199], respectively. By contrast, psoas muscle thickness had no predictive value [198]. However, in a subsequent cross-sectional study on 42 patients with LC [200], ultrasound muscle thickness did not offer an advantage over traditional bedside techniques, such as mid-upper arm muscle circumference (MUAMC) and bioelectrical impedance spectroscopy (BIS), when compared to a reference measurement of body cell mass derived from a multi-compartment model using isotope dilution tests and DXA [200].

Different studies have shown the prognostic value of muscle performance measures in LC, such as gait speed [201], the short physical performance battery [202] and the 6 min walking test [203]. Table 1 summarises the different methods available for sarcopenia screening and diagnosis.

### 3.4. Prevention and Treatment of Sarcopenia in NAFLD/Cirrhotic-NASH Patients

Since NAFLD and sarcopenia share risk factors and pathological mechanisms, a similar therapeutic choice based on lifestyle modification as a first-line approach is warranted. The treatment of the two pathologies must take place with a common approach based, first and foremost, on lifestyle modification.

#### 3.4.1. Diet

In overweight/obese NAFLD patients, moderate weight loss (<5%) is needed to improve hepatic steatosis; weight loss of 7–9% to improve histological features, such as steatosis, ballooning and inflammation; and weight loss >10% to improve fibrosis [204]. In lean patients with NAFLD, a modest weight loss of 3–5% is suggested, too [205]. The first treatment involves a hypocaloric diet (a daily reduction of 500–1000 kcal) associated with increased physical activity. The exclusion of NAFLD-promoting components (processed food and beverages high in added fructose) and a macronutrient composition adjusted according to the Mediterranean diet is also recommended. To prevent muscle mass loss, patients on the hypocaloric diet should receive a daily amount of protein of 1.2 g/kg/adjusted for ideal body weight [206], since lower protein intake has been related to a higher risk of sarcopenia [24], whereas higher protein intake in the context of energy restriction in individuals with obesity led to an attenuated loss of fat-free mass despite a more pronounced total body weight loss [207]. In NAFLD/NASH patients with sarcopenia, at least 1.2 and up to 1.5 g/kg actual body weight/day (ABW/d) dietary protein should be provided [206]. Protein provision of 1.2–1.4 g/kg ABW/d has been shown to prevent mass muscle reduction or increase muscle mass and strength in older and middle-aged women [208,209].

According to ESPEN guidelines, patients with LC should introduce 30–35 kcal/kg/day and 1.2–1.5 g/kg/day of proteins [204]. Guidelines also advise eating three meals and three snacks daily, including a late evening snack, to reduce night starvation [204].

#### 3.4.2. Exercise

It must be emphasised that exercise is essential for preventing and treating sarcopenia independently of weight loss [210,211]. Both aerobic and resistance training positively impact muscular mass, function and insulin sensitivity; however, resistance training is more effective in increasing muscle mass and strength [212], and it can attenuate the muscle mass and weight loss associated with calorie restriction [213,214]. On the other hand, being physically active contributes to the long-term maintenance of weight loss [215,216,217] and reduction in liver steatosis [218,219], even in the absence of weight loss [218]. Sarcopenia is inversely associated with physical activity levels in patients with NAFLD [25]. In a multicentre, retrospective study involving 11,690 NAFLD subjects, increased physical activity was significantly associated with a reduced risk of fibrosis, sarcopenia and ASCVD. In subjects with sarcopenic obesity or lean NAFLD, physical activity was also independently associated with a reduced risk of fibrosis and ASCVD [220]. In a systematic review and meta-analysis including seven RCTs enrolling patients with NAFLD and sarcopenia, physical function improved with endurance or combined training, but no evidence of improvement in muscle mass was seen. However, none of these studies evaluated muscle strength [221]. Therefore, even if epidemiological studies have confirmed an inverse association between physical activity and sarcopenia in patients with NAFLD, more studies are needed to delineate its role in improving muscle mass and strength.

Unlike non-cirrhotic NAFLD, the role of physical activity in improving body composition and physical performance in LC is emerging.

In a randomised clinical trial involving twenty-three cirrhotic patients, a 12-week exercise programme, compared to a relaxation programme, led to an increase in lean body mass assessed by DXA and an improvement in the cardiopulmonary exercise test and the Timed Up&Go test, whereas no changes were observed in the relaxation group [222]. Furthermore, in a randomised controlled trial including 39 outpatients with cirrhosis, patients who underwent 12 weeks of supervised progressive resistance training increased muscle strength and size and improved general performance measures compared with control subjects [223].

In a systematic review including 22 controlled trials and five single-arm interventions with small sample sizes (n = 6–120), the role of long-term (at least eight weeks) combined diet and exercise (high protein diets with aerobic and or resistance exercises) in improving body composition in LC patients has emerged. After reviewing eleven studies, the most effective physical activity protocol combines aerobic and resistance exercise with a minimum duration of 12 weeks, as noted by Williams et al. [224]. Work to 60–80% of maximal heart rate or VO2 peak is needed to significantly improve aerobic capacity [225,226,227]. Home-based physical programs can also improve physical performance, as shown by two small studies involving LC patients [228,229].

On the other hand, in a meta-analysis including four RCTs with 81 patients affected by end-stage liver disease, adapted physical activity proved safe, but no significant increase in muscle diameter, 6 min walking distance or VO2 peak changes were observed after exercise, probably due to the small sample size [230].

In conclusion, even if the existing literature has suggested the role of physical activity in improving sarcopenia in LC patients, additional studies with larger sample sizes are needed for confirmation.

#### 3.4.3. Supplementation

Branched-chain amino acids (BCAAs)—valine, leucine and isoleucine—are present in approximately 30–40% of muscle proteins and are involved in their synthesis; leucine especially stimulates protein synthesis through the mTOR pathway [231]. Furthermore, supplementation with BCAAs activates cell signalling pathways that increase myofibrillar protein synthesis [232]. Therefore, BCAA supplementation has been evaluated as a treatment for sarcopenia in LC patients. In a randomised, open-label, placebo-controlled study including 106 patients with LC and sarcopenia, treatment with BCAA for 24 weeks significantly improved muscle strength, function and muscle mass [233]. On the other hand, in a double-blind, randomised, placebo-controlled trial including 60 patients, BCAA, in addition to a home-based exercise program, did not significantly improve SMI, HSM or 6 m gait speed when compared to placebo [234].

A recent meta-analysis, including 11 studies (4 with an interventional study design, 5 with a prospective design and 5 with a retrospective design) and more than a thousand patients, evaluated the effect of BCAA supplementation on sarcopenia in LC patients [235]. The analysis revealed a significant increase in SMI and mid-arm muscle circumference (MAMC) when comparing baseline and post-intervention values. On the other hand, the increase in HGS and the decrease in triceps subcutaneous fat were insignificant. MAMC did not significantly increase when BCAA was compared to maltodextrin supplementation. Another recent systematic review, with only partial overlap of studies, highlights supplementary BCAAs when added to diet or exercise [236]. However, studies found high heterogeneity in BCAA dosage (6 to 110 g/day) and supplementation duration (3 to 12 months). Therefore, more research is needed to define the BCAA dosage and timing to prevent/treat sarcopenia in CLD patients, including the optimal combination of BCAA and resistance training to maximise the effect on muscle mass and function [237].

Beta-hydroxy-beta-methyl butyrate (HMB) is a leucine metabolite that can potentially increase muscle mass and performance by stimulating protein synthesis and reducing muscle catabolism. Two small, randomised trials exploring the role of HMB supplementation in LC patients led to conflicting results. In the first randomised, single-blind, placebo-controlled trial including 24 patients with LC, a dose of 3 g/day was administered for 12 weeks, with a significant increase in muscle function and quadriceps muscle mass measured by ultrasound and without adverse events [238]. In another randomised, controlled, double-blind trial including 43 subjects with LC, adding three g/day of HMB to the diet did not change fat-free mass, even if an upward trend in handgrip strength was observed [239]. Therefore, further data are needed to recommend HMB supplementation in LC.

Carnitine is a biofactor involved in fatty acid and energy metabolism, whose deficiency is frequently observed in patients with LC, especially in those with sarcopenia and malnutrition. Therefore, recent research has evaluated the role of L-carnitine supplementation in LC patients. Different studies have reported suppression of skeletal muscle loss [240,241,242] or increased skeleton [243] in patients with LC. This effect has been related to improving hyperammonemia [243]; yet, in other research, loss of skeletal muscle was significantly suppressed, even in patients receiving L-carnitine but not showing ammonia decrease [241]. In a study enrolling 18 patients with LC, L-carnitine supplementation (1000 mg/day) associated with BCAAs and additional exercise for six months did not improve HGS but reduced the frequency of complaints of muscle cramping [244]. Different dosages of L-carnitine have been used, ranging from 1000 to 4000 mg/day [240,241,242,243,244], with treatment durations from 3 to 12 months [240,241,242,243,244]. Hence, even if L-carnitine has a potential therapeutic role in sarcopenia in LC, more research is needed to determine whether adding carnitine to a dietary intervention, including BCAAs, is necessary and at what dosage.

Low vitamin D levels have been reported [245,246] and identified as an independent factor for sarcopenia in patients with CLD [247,248]. Therefore, a randomised controlled study enrolling 33 patients with decompensated LC has explored its role as a therapeutic option for sarcopenia. Those who received oral native vitamin D3 at a dose of 2000 IU once a day for 12 months had a significantly greater increase in SMI (median change rate +5.8%) and grip at 12 months (*p* = 2.57 × 10^−3^ and 9.07 × 10^−3^), with a significant decrease in the prevalence of sarcopenia from 80.0% (12/15) to 33.3% (5/15; *p* = 2.53 × 10^−2^) [249]. These results need to be confirmed by further studies.

#### 3.4.4. Pharmacotherapy

Since testosterone treatment in older adults was associated with increased muscle mass [250] and improved physical performance [251,252], this hormonal therapy has been evaluated to treat sarcopenia in LC patients. In a 12-month, double-blinded, placebo-controlled trial including 101 men with LC, intramuscular testosterone led to higher appendicular and total lean mass and reduced fat mass without an increase in adverse events [253]. The beneficial role of this treatment has to be assessed in larger cohorts of patients before entering clinical practice.

Metformin is an insulin sensitiser drug widely used for treating type 2 diabetes; it is safe for the liver but not specifically recommended to treat NAFLD [254,255]. Currently, it is being investigated as a potential anti-sarcopenic agent. In fact, it has an inhibiting effect on NF-kappaB-mediated inflammation and oxidation response; it can mimic exercise-activated activated protein kinase (AMPK) signalling, which stimulates muscle anabolism. However, this latter represents a dose–response phenomenon, characterised by low-dose stimulation and high-dose inhibition of AMPK signalling, which raises concerns about a possible untoward effect opposite to the expected one [256]. There are not enough data to establish a favourable, neutral or unfavourable clinical effect of this drug on sarcopenia.

Several novel drugs are currently being investigated for sarcopenia treatment, targeting multiple pathways, such as myostatin, the renin-angiotensin system, androgen receptors, activated protein kinase and ghrelin [256]. However, no published data on these drug trials on liver disease-associated sarcopenia are available. Table 2 summarizes the available evidence on supplementation and pharmacotherapy in the prevention and treatment of sarcopenia in CLD.

## 4. Bone Density Defects

### 4.1. Osteoporosis and Bone Density Impairment in NAFLD

Osteoporosis (porous bone) is a disease characterised by bone weakness and an increased risk of fragility fractures that can occur as a result of low-level trauma. A fragile bone can be easily damaged by impact or stress that would not hurt a bone with a normal mineral density [257]. Bone’s ability to withstand trauma is the result of a combination of quality elements (structure, micro-architecture and mineral composition of the bone tissue) and quantitative factors (i.e., bone mineral density (BMD)) [258]. BMD is a measurable parameter consisting of the amount of bone mass per unit area or unit volume [259]. It is crucial for diagnosing bone impairment and provides prognostic information for assessing fracture risk.

Osteoporosis is more common among the elderly population because the bones become more porous and weak with age; osteoporosis is also more frequent among women than men because of the reduction in circulating oestrogens following menopause [260]; the morbidity of osteoporosis comes from the associated fractures that significantly impact overall survival [260].

Moreover, even though it appears as a steady element, bone is an active tissue with a high turnover rate: tissue is incessantly resorbed by osteoclasts and rebuilt by osteoblasts [261]. This process is controlled by local signals (such as growth factors and cytokines) and systemic signals (parathyroid hormone, calcitonin and oestrogen), which cooperate to maintain bone homeostasis [262]. In the elderly, the unbalance between bone formation by osteoblasts and bone resorption by osteoclasts is crucial in the pathogenesis of bone density defects.

Considering the role of the liver in bone metabolism, especially in the vitamin D pathway, the link between NAFLD and bone defects is not surprising [263]. Vitamin D is a fat-soluble vitamin involved in calcium/phosphorus metabolism and bone homeostasis. The main form of vitamin D is vitamin D3 (cholecalciferol) which originates from 7-dehydrocholesterol after ultraviolet (Uv) light irradiation. Vitamin D’s principal circulating and storing metabolite is 25-hydroxyvitamin D [25(oH)d], produced in the liver after the 25-hydroxylation of Vitamin D by 25-hydroxylase. Subsequently, 25(oH)d is converted to 1.25(OH)2D in the kidney by 1α-hydroxylase. The biological actions of vitamin D are performed after the binding of 1.25(oH)2d to its receptor. When its ligand binds it, it directly or indirectly leads to the expression of various genes, which impact cell differentiation, proliferation, apoptosis, angiogenesis and immunomodulation. In addition, low serum 25(oH)d levels have been reported to be connected to components of the metabolic syndrome [264]. Significant evidence suggests an association between low serum 25(OH)D and NAFLD [265]; hypovitaminosis D is firmly and independently related to NAFLD in the normal adult population without altered liver enzymes [266].

### 4.2. Prevalence of Bone Density Defects in NAFLD, Cirrhotic NASH and Prognosis Implication

Multiple cross-sectional studies have attempted to clarify the association between NAFLD and bone density defects [267]. Understandably, many conflicting data have been published because of various confounding factors, such as age, sex, race, and nutritional and menstrual conditions, which could influence bone metabolism.

A few studies have ascertained a notable link between lower BMD or higher prevalence of fragility fractures in adolescents and adults and NAFLD [268,269]; however, some studies identified no relevant associations between the two conditions [270,271]. Furthermore, it has also been noticed that there is no association between NAFLD and reduced BMD [272,273], which may suggest that NAFLD (especially NASH) could worsen insulin resistance and determine the release of several pro-inflammatory cytokines and bone-affecting molecules with the development of osteoporosis [274].

In a recent study, Pan et al. evaluated the association between the prevalence and risk of osteoporosis or osteoporotic fracture with NAFLD. The results showed that the prevalence of osteoporosis was higher in male and female NAFLD groups than in the non-NAFLD group, and the risk of osteoporosis or fragility fractures was higher in the fatty liver disease group than in the non-NAFLD group. Finally, data indicated that the risk of osteoporosis or osteoporotic fractures was higher in the male NAFLD group than in the non-NAFLD group, without significant differences among women [274]. Moreover, evidence suggests an increased fracture risk in individuals affected by NAFLD with liver fibrosis compared to NAFLD controls without fibrosis. This could be explained by the reduced liver synthesis of insulin-like growth factor-1 (IGF-1), which stimulates osteoblast proliferation and differentiation resulting in a lower bone turnover [275]. In addition, the surplus of lipid accumulation in the liver determines low-grade, chronic inflammation, which is combined with the development of bone loss, osteoporosis and a higher risk of osteoporotic fractures; accordingly, pro-inflammatory cytokines have an osteoclastogenic activity [276]. For instance, TNF-α can decrease trabecular and cortical bone formation by inhibiting osteoblast differentiation and encouraging osteoblast apoptosis. This cytokine can also promote trabecular and cortical bone resorption by determining the expression of the receptor activator of the nuclear factor kappa-B ligand (RANKL), which prevents osteoclast activation and osteoblast apoptosis [277].

Zhai et al. evaluated the clinical implications of bone impairment in 13,837 adults with NAFLD in the United States, resulting, after adjustment for potential confounding factors, in an increase in the prevalence of osteopenia/osteoporosis in the femoral neck in adults aged ≥ 40 years throughout 2005–2014; furthermore, NAFLD complicated with fibrosis was positively associated with the occurrence of spine fracture [278]. Moreover, hip fractures are firmly associated with reduced BMD at the femoral neck. Hip fractures, which present with pain and an inability to bear weight, almost always demand surgical correction and are combined with a greater reduction in functional status and quality of life than all other types of fracture, with a high risk for short-term mortality [262]. Despite no studies having assessed the relationship between osteoporosis or fragility fractures and survival in patients with NAFLD (or other chronic liver diseases), it is possible to extrapolate some evidence from studies conducted in the general population. Using the DALY (“disability-adjusted life year”) as a measure of burden disease, according to the WHO’s standards, it has been evaluated that fragility fractures are the fourth most severe illness, preceded only by ischemic heart disease, dementia and lung cancer [260]. It is reasonable to assume that fragility fractures should represent a major issue in patients with chronic liver disease, considering their remarkable clinical frailty, and they probably could be a negative prognostic factor in terms of quality of life and survival.

### 4.3. Diagnosis and Clinical Assessment of Bone Density Defects in NAFLD

The parameter used to identify bone defects is the BMD measure, detected with dual-energy X-ray absorptiometry (DXA). BMD values are standard deviations from the young population’s mean values for the diagnosis. This parameter is called the T-score, and the measurement of the T-score can lead to the diagnosis of osteoporosis (T score ≤ −2.5) or osteopenia (−2.5 < T-score ≤ −1.0) [261].

The bone density of the entire skeleton can be examined, but the most commonly evaluated points are the hip (femoral neck), lumbar region (L1–L4) and wrist. The lifetime risk for femoral, vertebral and wrist fractures is around 40%, similar to cardiovascular disease [257]. Evidence indicated that fragility fractures of osteoporosis were 2.5-fold more common in subjects with NAFLD [279,280]; furthermore, low BMD was more relevant in patients with progressive liver disease, including NASH, and significant fibrosis [279,281]. Intriguingly, Ahn et al. reported that fatty liver index (FLI), a composed formula used to predict the presence of NAFLD (BMI, waist circumference, GGT and triglycerides), was negatively associated with BMD (evaluated with DXA) at all skeletal sites only in men [282]. The link between FLI, BMD and the risk of osteopenia and osteoporosis was independent of insulin resistance [282]. Moreover, diagnosing, monitoring and eventually supplementing with vitamin D and calcium is mandatory in patients with osteopenia or osteoporosis for the known positive effects on bone metabolism [283].

### 4.4. Prevention and Treatment of Bone Defects in NAFLD/Cirrhotic-NASH Patients

Lifestyle modification is crucial in preventing and treating NAFLD [284]; weight loss is connected with reducing liver fat accumulation and levels of aminotransferase [285]. The extent of weight loss is a determinant of histological improvements in NAFLD and NASH. Although small reductions (3−5% body weight loss) can improve hepatic steatosis and the associated metabolic indicators, superior weight loss (at least 7%) is necessary to improve or resolve NASH [286,287].

Following a Mediterranean diet, with lower calories than required daily, is associated with reduced body weight, hepatic lipid accumulation and insulin resistance [288].

Nevertheless, physical exercise in NAFLD and chronic liver disease is suggested to improve liver disease and maintain and improve BMD and skeletal muscle [289] since physical stress positively impacts bone formation [290]. A previous study evidenced the positive effect of weight-bearing aerobic exercise and resistance training on BMD, and exercise could reduce the risk of falling and fractures in patients affected by osteoporosis [290].

Fragility fractures of osteoporosis can be prevented and treated with several medications, which can be divided into two main categories of drugs: anti-resorptives (mainly bisphosphonates and denosumab) and anabolics (teriparatide and abaloparatide) [262].

Bisphosphonates (BPs) are the most widely used drugs to prevent and treat osteoporosis and osteoporotic fractures. Their ability to be incorporated into the bone matrix, related to their structural analogy to inorganic pyrophosphate, accounts for their high specificity of action expressed only at the bone level, thus reducing pharmacological interactions with other medications and possible side effects [291].

Unfortunately, there have been no large clinical trials specially carried out to assess bisphosphonates’ impact on bone impairment in NAFLD/NASH/CLD, but there are a few trials that have been conducted to evaluate the possible efficacy of BPs in patients with primary biliary cholangitis and secondary osteoporosis. Just one RCT performed to compare BPs and a placebo demonstrated improvement in spine and femoral BMD [292]. In 2005, Zein et al. compared the oral administration of alendronate with a placebo in a group of 34 participants with PBC. After one year, a statistically significant improvement in lumbar spine BMD and proximal femur BMD from the baseline was observed in the alendronate group compared to the placebo group, while no difference was observed in the number of vertebral and non-vertebral fractures [293]. Other studies have been performed to compare the use of different BPs in bone loss following PBC [294,295], but not one has shown any effect in reducing the risk of fracture [292]. A lack of evidence may be found in an Indian study, which evaluated the impact of BP administration on a group of cirrhotic patients (notice that cholestatic liver disease was an exclusion criterion) [296]. Bansal et al. orally administered ibandronic acid to cirrhotic patients with osteoporosis, and after six months of follow-up they reported a significant improvement in lumbar and femoral BMD. No effect on fractures has been observed. This study provides low-quality evidence, considering that only 40% of patients completed the trial and were followed up for six months.

The main concern about the oral administration of bisphosphonates in patients with liver disease is the risk of gastro-oesophageal erosion, especially in patients with varices [297]. Although this issue could be avoided through the intravenous administration of bisphosphonates, in none of the analysed studies have serious adverse events been reported [292,293,294,295,296]. Two non-randomised controlled trials have been performed to evaluate the safety of risedronate oral administration in patients with liver cirrhosis and oesophageal varices [298,299]. These studies confirm the effectiveness of BPs in increasing bone mass, especially at the lumbar level. In fact, after one year of treatment, the intervention group showed a significant improvement in lumbar BMD compared to the control group. This improvement was even higher after two years, showing that BMD continued to increase. No improvement in femoral BMD has been noticed. Despite the drug’s increased risk of oesophagitis, no gastrointestinal bleeding occurred. It must be noticed that the studies enrolled only patients with low-risk varices, while patients with high-risk bleeding varices underwent pre-intervention EVBL. This point prevents a generalisation about BP administration in cirrhotic patients. Still, it suggests that these medications can be used safely in patients with a low risk of bleeding varices. In contrast, greater caution is required in patients with a higher risk of bleeding.

Denosumab is a human monoclonal IgG2 antibody capable of linking with high affinity to RANKL, a key cytokine in stimulating bone resorption. It inhibits osteoclast activation and function by preventing its interaction with the RANK receptor, expressed on the osteoclast’s surface [300]. Compared to bisphosphonates in common clinical practice, denosumab has some main advantages: (1) it leads to a greater improvement in all-site-BMD than BP [301]; (2) denosumab involves a semestral parenteral administration (subcutaneous injection), and it should provide better patient adherence instead of the weekly oral administration of BP; (3) no studies have been performed to assess the safety of BP administration in patients with severe renal impairment, so BP use is contraindicated in subjects with eGFR < 30 mL/min, as opposed to denosumab. Only a retrospective study observed denosumab administration in chronic liver disease patients. It showed a significant improvement in all BMD values and no reported fractures or mild/severe adverse events [302]. This study has many limitations: first of all, it involved only 60 patients with a very short follow-up of just one year. Despite these limits, it provides evidence of denosumab’s effectiveness in reducing bone loss in chronic liver disease patients.

Partial evidence reported that RANKL upregulation could play a key role in NAFLD pathogenesis and be the possible link between hepatic and bone impairment, so its downregulation could be protective for liver health (and bone, too) [303,304]. A suggestive paper reported amelioration of liver function in a woman with NASH and osteoporosis after denosumab administration [305].

While anti-resorptive drugs improve bone mineral density inhibiting osteoclast activity and preventing loss of bone tissue, anabolic medications stimulate the production of new bone tissue. The main example of this class is teriparatide, an analogue of the human parathyroid hormone (1–34). Despite its effectiveness in gaining bone mass and consequently improving BMD, especially at the spine, this medication cannot be administered for long periods because an increased incidence of osteosarcoma in murine models treated with high doses of teriparatide has been noticed [306]. For this reason, many regulatory agencies have established a maximum treatment period with teriparatide for up to two years. The other limit of anabolic therapies is that their effect on BMD is transient, so at the end of treatment, a switch to anti-resorptive medications is needed to avoid a rebound of bone tissue loss. There are no studies assessing the safety and efficacy of anabolic medications in patients with NAFLD/CLD/ESLD and osteoporosis, but they are probably not suitable for this type of patient, especially regarding the cancer risk.

Despite the high prevalence of NAFLD, no efficacious drugs are currently approved for treating the disease; among the drugs proposed, pioglitazone is used off-label in selected patients affected by NASH with fibrosis stage ≥ 2 and type 2 diabetes [307,308]. However, pioglitazone appears to be linked to an increased risk of osteoporotic fractures [309,310]. Since NAFLD and osteoporosis share some molecular and biological pathways, employing drugs that improve both conditions may be important. Recently, GLP1-RAs (mainly Semaglutide and Liraglutide), a class of anti-diabetic and anti-obesity drugs, have provided intriguing data, showing a reduction in liver steatosis and inflammation [311,312], and, at the same time, they appear to have an impact on bone metabolism [310], with favourable effects on bone mineral density [313] and reduction in the risk of fractures [314].

## 5. Conclusions

Patients with poor bone mineral density and sarcopenia are referred to as having osteosarcopenia. Due to the additional negative health consequences that the two diseases predispose towards, those with chronic liver disease and osteosarcopenia are at an increased risk (Figure 1). Osteosarcopenia consequences have a large cost impact on health systems because they are linked to considerable increases in morbidity and mortality.

Early detection of the condition, either by screening people at higher risk, as seen above, or by evaluating patients for clinical characteristics of osteosarcopenia, may assist in avoiding negative effects. Patients with chronic dysmetabolic liver disease may undergo a significant decrease in falls, fractures and functional impairment when following straightforward therapies, such as resistance exercise, appropriate dietary protein and calcium consumption, and maintenance of normal vitamin D levels.

## Figures and Tables

**Figure 1 ijms-24-07517-f001:**
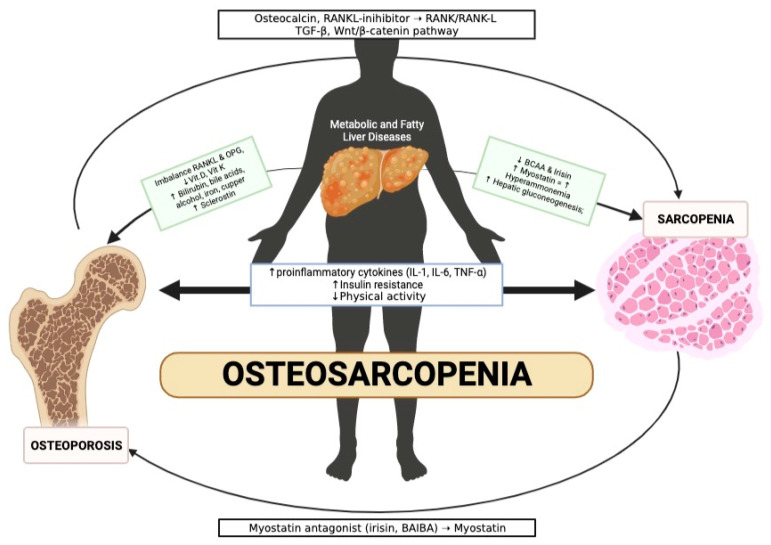
Pathophysiological mechanism and the association between osteoporosis and sarcopenia in non-alcoholic fatty liver disease.

**Table 2 ijms-24-07517-t002:** Supplementations and drugs for prevention and treatment of sarcopenia in LC patients.

Molecule	Mechanism	Clinical Effects	Type of Evidence	Limitations
BCAAs	Increased myofibrillar protein synthesis through the mTOR pathway	Significant increase in SMI and MAMC Insignificant increase in HGS	Meta-analysis, including 11 studies	High heterogeneity in dosage (6 to 110 g/day) and supplementation duration (3 to 12 months) Timing of supplementation and combination with resistance training to be defined
HMB	Stimulation of protein synthesis	Conflicting results: significant increase in muscle function and quadriceps muscle mass vs. absence of change in FFM with un upward trend in HGS	Small RCTs	Larger studies are needed to assess efficacy
L-carnitine	Involved in fatty acid and energy metabolism	Suppression of skeletal muscle loss	A few RCTs and retrospective studies	Heterogeneity of dosages (from 1000 to 4000 mg/day) and treatment durations (from 3 to 12 months)
Vitamin-D	Acts through vitamin D receptor in muscles, but detailed mechanisms are still unclear	Increase in SMI	A small RCT	More studies needed to confirm the beneficial effect
Testosterone	Stimulates muscle protein synthesis	Higher appendicular and total lean mass and reduced fat mass	An RCT	Assessment in larger cohorts needed
Metformin	It can mimic exercise-activated activated protein kinase (AMPK) signalling, which stimulates muscle anabolism	NA	NA	Clinical effect on sarcopenia needs to be evaluated

BCAAs, branched-chain amino acids; FFM, fat-free mass; HMB, beta-hydroxy-beta-methyl butyrate; HGS, handgrip strength; MAMC, mid-arm muscle circumference; NA, not available; RCTs, randomised controlled trials; SMI, skeletal muscle index.

## Data Availability

Not applicable.

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
