# Peer review of "Osteosarcopenia in NAFLD/MAFLD: An Underappreciated Clinical Problem in Chronic Liver Disease"

_ijms, 2023, doi:10.3390/ijms24087517_

Round 1
Reviewer 1 Report
In the review entitled "Osteosarcopenia in NAFLD/MAFLD: an underappreciated clinical problem in chronic liver disease”. The authors put in evidence the relationship between osteosarcopenia and bone impairment in NAFLD/MAFLD patients, highlighting the main strategy used for the early identification and management of these patients. Nevertheless, there are some suggestions to improve the manuscript.
1. The abstract is very short; it should be developed more, respecting the instructions of the authors in the International Journal of Molecular Sciences.
2. The reference numbers should be placed in square brackets []. It recommended preparing the references with a bibliography software package, such as EndNote, ReferenceManager or Zotero. Please use the International Journal of Molecular Sciences guide for authors and the reference list and citations guide to improve them.
3. On page 7 line 33, the phrase “remarkable ……and survival” is written in Italic. It should be written in the same typeface as the paragraph.
4. On page 9 line 21, the phrase “has been conducted” is crossed out. Please correct it.
5. In the paragraph under title : 3.3.Diagnosis and clinical assessment of sarcopenia in NAFLD and LC on line 40 the defined article “the” was written with a larger font size. Please correct it.
6. The Author Contributions section are missing from the manuscript. Should be added.
In the section of the abstract:
p. 1, line 4; mass losses → mass loss
In the section of the introduction:
p. 1, line 1; recently also → also recently
p. 1, line 8; NAFLD increased → NAFLD has increased
p. 1, line 9; By histological findings → From histological finding; lifestyle, overweight → lifestyle, being overweight
p. 1, line 13; worsen and become → worsen and cause
p. 1, line 17; following small→ following a small
In the paragraph under title: 3.1. Definition of sarcopenia (S) and sarcopenic obesity (SO)
p. 2, line 17; be gold → be the gold
p. 2, line 19; and-to → and to; extent- Bioelectrical → extent Bioelectrical
p. 2, line 38; when skeletal → when the skeletal
p. 2, line 44; obesity has been → obesity was
p. 2, line 52; NAFLD and with → NAFLD, with
In the paragraph under title: 3.2.1. Sarcopenia and NAFLD
p. 4, line 13; particularly android → particularly the android
p. 4, lines 13-14; and android-fat-to-gynoid-fat → and the android to-gynoid-fat
p. 4, line 21; and higher BMI → and a higher BMI
p. 4, line 25; of absolute → of an absolute
p. 4, line 34; this was available only in → this is available in only
p. 4, line 40; (AMC) - defined → (AMC) defined
p. 4, line 41; 2.1â‚“ and 3.3â‚“ higher → 2.1-fold and 3.3-fold higher
In the paragraph under title: 3.2.2. Sarcopenia and NASH liver cirrhosis (LC)
p. 5, line 1; from different → of different
p. 5, line 8; with third → with the third
p. 5, line 10; 20 studies (7 studies Asian and 13 studies western) → 20 studies (7 Asian and 13 western)
p. 5, line 20; increased risk of mortality and reduced survival → increased mortality, as well as reduced survival
In the paragraph under title: 3.2.3. Sarcopenia and liver transplant (LT)
p. 5, lines 2-3; ranging from 22% and 70% → ranging from 22% to 70%
p. 5, lines 3-4; with WL and post-LT and with post-LT → with WL, post-LT and post-LT
p. 5, line 6; from different etiologies → of different etiologies
In the paragraph under title: 3.3. Diagnosis and clinical assessment of sarcopenia in NAFLD and LC
p. 1, lines 2-3; and calf → and the calf
p. 1, line 5; it resulted → it results
p. 1, line 14; is a gold → is the gold
p. 1, line 25; not extend → not apply
p. 1, line 33; with the cut-off → with a cut-off
p. 2, line 54; with an excellent → with excellent
p. 2, line 75; and correlates → which correlates
In the paragraph under title: 3.4.1. Diet
p. 3, line 5; treatment includes → treatment involves
p. 3, line 7; food, d food and beverages → food and beverages
In the paragraph under title: 3.4.2. Exercise
p. 3, line 2; independently from weight → independently of weight
p. 3, line 4; ,ae, it can→ it can
p. 3, line 5; with caloric restriction → with calorie restriction
p. 4, line 28; and benefitted → and improved
p. 4, line 35; of exercise program of 12 weeks → of 12 weeks
p. 4, line 40; affected with → affected by
In the paragraph under title: 3.4.3. Supplementation
p. 4, line 3; muscle protein through → protein synthesis through
p. 4, line 19; partial overlapping → partial overlap
p. 4, line 20; highlights supplementary → highlights the benefit of supplementary
p. 4, line 25; mass and unction → mass and function
p. 4, line 26; metabolites and can → metabolites that can
p. 4, line 30; a 3g/day → a dose of 3g/day
In the paragraph under title: 3.4.4. Pharmacotherapy
p. 5, line 15; which raises concern → which raises concerns
p. 5, line 9; which is safe → it is safe
In the paragraph under title: 4.1. Osteoporosis and bone density impairment in NAFLD
p. 6, lines 3-4; and menstrual condition → and menstrual conditions
p. 6, line 10; because the bone becomes → because the bones become
p. 6, lines 11-12; than men since → than men because of
p. 6, line 13; the morbidness of → the morbidity
p. 6, lines 13-14; significantly to overall → significantly on the overall
p. 6, line 14; was increased in both → was higher in male and female
p. 6, lines 15-16; a high turnover → a high turnover rate
p. 6, line 31; Also, low → Also, a low
p. 6, line 32; components of metabolic → components of the metabolic
p. 7, line 24; inflammation, combined → inflammation which is combined
In the paragraph under title: 4.2. Prevalence of bone density defects in NAFLD, cirrhotic NASH and prognosis implication
p. 7, line 20; compared to the NAFLD → compared to NAFLD
p. 7, line 22; stimulates osteoblasts → stimulates osteoblast
p. 7, line 34, 37; at the femoral neck → in the femoral neck
p. 7, line 44; as a meter of → as a measure of
In the paragraph under title: 4.4. Prevention and treatment of bone defects in NAFLD/cirrhotic-NASH patients
p. 8, line 35; administration in a group → administration on a group
p. 8, line 43; gastroesophageal erosions → gastroesophageal erosion
p. 9, line 52; BMD continued to increase → BMD has continued to increase
p. 9, line 54; it must be notice → it must be noticed
p. 9, line 80; inhibiting osteoclasts activity → inhibiting osteoclast activity
Reviewer 2 Report
This review made a comprehensive and detailed summary focused on osteosarcopenia and NAFLD. It highlights the weakness and problems on assessing sarcopenia in patients with NAFLD. The authors have provided sufficient evidence demonstrating the underappreciation of osteosarcopenia and the incomplete studies in this field. However, there are some questions need to be answered in this review:
1. In the introduction, the authors mentioned the common pathophysiological pathways shared by osteoporosis and NAFLD. If the mechanisms are listed in Figure 1, do the prevention and treatment approaches (in 3.4 and 4.4) function through these shared pathways? If yes, please specify some drugs or pathways in the introduction.
2. There is a table for the methods of assessment of sarcopenia. It is better to have a similar table summarizing the limitations of supplementation or pharmacotherapy in the section of prevention and treatment.
3. Are there biomarkers identified for diagnosis of osteosarcopenia or NAFLD? Have they been used for clinical assessment or predicting the risk of chronic liver disease?
4. In the summary Figure 1, it is better to propose a clearer model that demonstrates association between scarcopenia/osteoporosis and liver disease. This review is trying to summarize the impact of sarcopenia and bone impairment on NAFLD/MAFLD whereas Figure 1 only demonstrates the risk factors caused by liver disease.
